# ON THE ROLE OF RIEMANNIAN METRIC IN ISOMETRIC REPRESENTATION LEARNING

## ABSTRACT

Under the manifold hypothesis, isometric representation learning aims to discover a set of latent space coordinates that preserve the manifold's geometry. The geometry of the manifold needs to be specified prior, and typically, it is defined with a metric inherited from the ambient data space. Most existing methods adopt the identity metric assumption of the ambient space (namely, Euclidean data space), a choice that is likely one of the most reasonable in unsupervised contexts. However, this unsupervised selection of the identity metric inherently lacks the capacity to capture the semantic understanding that humans perceive from data. The question of how to formulate a data-semantic-aware Riemannian metric for the ambient space remains unanswered, particularly in the context of isometric representation learning. In this work, we propose a method for constructing *neural feature-based metrics* capable of capturing data semantics by adopting knowledge from any pre-trained feature extraction model. Then we conduct a comparative study on the effects of the following Riemannian metrics in isometric representation learning: (i) the identity metric, (ii) the inverse density-based metric – which is an existing unsupervised metric construction method –, and (iii) the proposed neural feature-based metrics. Experiments with standard image datasets *MNIST*, *Fashion MNIST*, and *CIFAR10* show that the neural feature-based metrics produce data-semantic-aware representations – where data with similar semantics are located nearby – and in some cases are able to discover unseen hierarchical structures in the datasets.

## 1 INTRODUCTION

Under the manifold hypothesis (Hastie & Stuetzle, 1989; Smola et al., 2001) – that is, a set of data points drawn from some high-dimensional space $\mathcal{X}$ are assumed to lie approximately on some lower dimensional manifold $\mathcal{M} \subset \mathcal{X}$ –, the *manifold representation learning* aims to find the manifold and its coordinate space $\mathcal{Z}$ (or the *latent* space) via a non-linear mapping $f : \mathcal{Z} \to \mathcal{X}$ such that $f(\mathcal{Z})$ closely approximates $\mathcal{M}$. Using sufficiently flexible deep neural network models for $f$, the autoencoder (Kramer, 1991) and its variants (Kingma & Welling, 2013; Vincent et al., 2010; Rifai et al., 2011; Tolstikhin et al., 2017; Lee et al., 2021; Jang et al., 2023; Lee & Park, 2023) offer an appealing framework to learn manifold representations.

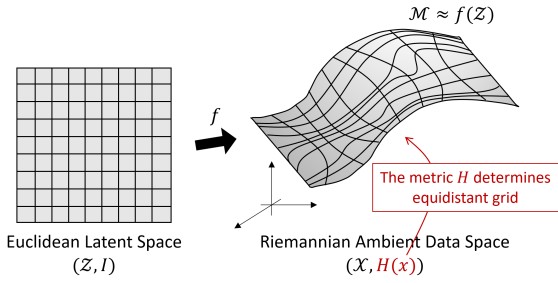

Figure 1: The Riemannian metric $H(x)$ in the ambient data space determines the geometry of the manifold $\mathcal{M}$; a geometry-preserving map $f$ maps the Euclidean equidistant grid in $\mathcal{Z}$ to the equidistant grid in $\mathcal{M}$.

Unfortunately, the latent spaces of autoencoders are often distorted and fail to preserve the manifold's geometric structure. For instance, straight lines in the latent space do not correspond to *geodesics*, which are the shortest curves along the manifold (Arvanitidis et al., 2018; Shao et al., 2018). To address this issue, one approach involves constructing the Riemannian metric for the latent space by pulling back the data space metric via $f$. Geodesics can then be obtained by solving

the geodesic equations (Arvanitidis et al., 2018; Shao et al., 2018; Yang et al., 2018). However, this process usually demands solving a computationally intensive boundary value problem, leading to the development of fast and approximate geodesic computation methods (Arvanitidis et al., 2019; Chen et al., 2019).

Recently, regularization methods that enforce the learning of geometry-preserving mappings $f$ have been developed, to ensure that the latent space Euclidean geometry closely approximates the geometry of the underlying data manifold (Chen et al., 2020a; Lee et al., 2022b; Nazari et al., 2023). There is a hierarchy of geometry-preserving mappings (Lee et al., 2022b), where at the top of the hierarchy are isometries that preserve distances and angles, followed by scaled isometries that preserve angles and scaled distances, and then angle-preserving and area-preserving maps. The mapping $f$ is regularized to be a scaled isometry in (Lee et al., 2022b) and produces a near *isometric representation space* $\mathcal{Z}$ – where the straight lines in $\mathcal{Z}$ approximate geodesics in the manifold.

The isometric representation learning in (Lee et al., 2022b) proceeds with the following two steps: (i) constructing a Riemannian metric for the ambient data space $\mathcal{X}$ while assuming the identity metric for the latent space $\mathcal{Z}$ and (ii) finding $f : \mathcal{Z} \to \mathcal{X}$ that parametrizes the manifold $\mathcal{M}$ while being close to a scaled isometry between the two spaces (see Figure 1). However, existing studies of geometry-preserving maps are limited to assuming the identity metric for the ambient space $\mathcal{X}$, with the exception of (Lee et al., 2022a) which designed a specialized metric for point cloud data. In a general sense, the consequences of selecting different Riemannian metrics in isometric representation learning have not been investigated thus far.

The study on the Riemannian metric for data space has its own history. One of the earliest works suggested a convex combination of a predefined set of local metrics, each of which is learned using labeled data (Hauberg et al., 2012). In an unsupervised setting, Arvanitidis et al. (2016) constructed the metric in a non-parametric manner as the inverse of the local diagonal covariance, however, it requires tedious parameter tuning. More recently, in (Arvanitidis et al., 2020), an unsupervised method that constructs a diagonal metric as the inverse density has been proposed, where the density is learned with a parametric model, e.g., GMM (Reynolds et al., 2009).

In this paper, we take the initial stride toward identifying the role of data space Riemannian metric in isometric representation learning. First of all, we use one of the unsupervised Riemannian metric construction methods, the *inverse density-based metric* from (Arvanitidis et al., 2020). This construction assigns large volumes to low-density regions and small volumes to high-density regions; therefore, it is naturally expected that, in isometric representations, the low-density regions are expanded while high-density regions are contracted, leading to a clearer clustering structure.

Unsupervised techniques for metric construction however can only grasp density-related information but fundamentally fall short of capturing the semantic understanding that humans perceive from data. We propose a *neural feature-based metric* that captures data semantics, by leveraging semantic knowledge transferred from any pre-trained neural feature extraction model. In this work, we consider self-supervised and transfer learning scenarios via the following two types of neural feature extraction models: (i) models trained via self-supervised learning by exploiting a set of pre-specified semantic-preserving transformations (Chen et al., 2020b; Tian et al., 2020; Caron et al., 2020; Grill et al., 2020; Bardes et al., 2021) and (ii) models pre-trained on large-scale datasets capable of extracting data-semantic-relevant features (Simonyan & Zisserman, 2014; Szegedy et al., 2016; Deng et al., 2009; Heusel et al., 2017; Raghu et al., 2021).

Experiments are conducted with standard benchmark image datasets such as the *MNIST* (Deng, 2012), *Fashion MNIST* (Xiao et al., 2017), and *CIFAR10* (Krizhevsky et al.). We use some well-known image transformations (e.g., flip, crop and resize) for self-supervised neural feature-based metrics, and use classifiers trained on ImageNet (Deng et al., 2009) as large-scale pre-trained models for transfer learning. Both qualitative and quantitative analyses are given in two-, three–, and higher-dimensional latent spaces. Compared to the identity metric choice, the invden metric produces a more apparent clustering structure, by exploiting density information. The neural feature-based metrics produce more semantic-aware representations where data with similar semantics are located nearby, supported by the increased accuracy of latent KNN classification. Moreover, in certain instances, the neural feature-based metrics are able to discover unseen hierarchical structures within the data distribution that are pertinent to visual appearance, offering a more detailed understanding of the dataset.

## 2 ISOMETRIC REPRESENTATION LEARNING

This section briefly introduces the isometric regularization method for autoencoders proposed in (Lee et al., 2022b); the details regarding a hierarchy of geometry-preserving mappings, general expressions for coordinate-invariant functionals, and distortion measures can be found in the original paper. Throughout, we assume that the data space $\mathcal{X} = \mathbb{R}^D$ is assigned the Riemannian metric $H(x) \in \mathbb{R}^{D \times D}$ for $x \in \mathcal{X}$ that is a positive-definite matrix; the construction of this metric is discussed in the next section. Assuming the underlying manifold dimension is $m < D$, the latent space is set as $\mathcal{Z} = \mathbb{R}^m$ and is assigned the identity metric $I_m \in \mathbb{R}^{m \times m}$ for all $z \in \mathcal{Z}$.

An autoencoder is defined as a pair of mappings, an encoder $g : \mathcal{X} \to \mathcal{Z}$ and a decoder $f : \mathcal{Z} \to \mathcal{X}$; we denote the Jacobian of $f$ by $J_f(z) := \frac{\partial f}{\partial z}(z) \in \mathbb{R}^{D \times m}$. It is important to note that the image of $f$ only occupies a lower-dimensional area in $\mathcal{X}$, that is an $m$-dimensional manifold under some mild conditions[1]. This $m$-dimensional manifold parametrized by $f$ will be denoted by $\mathcal{M} = f(\mathcal{Z})$ and is embedded in the ambient space $\mathcal{X}$. We consider the projection of the ambient space metric $H(x)$ to the manifold $\mathcal{M}$ as the Riemannian metric for $\mathcal{M}$. Given the projection metric, the inner product between two tangent vectors $v, w \in T_x \mathcal{M}$ is defined by using the ambient space metric as $\langle v, w \rangle_x := v^T H(x) w$. As a direct consequence, given a curve in the manifold $x(t) \in \mathcal{M}$, the velocity norm of the curve is defined as $\dot{x}^T H(x) \dot{x}$.

To gain an intuition for the definitions of the scaled isometry and relaxed distortion measure used in isometric regularization, consider a curve $z(t) \in \mathcal{Z}$ for $t \in [0, 1]$ in the latent space and its corresponding curve in the manifold $x(t) := f(z(t)) \in \mathcal{M}$. Then, the length of the curve $x(t)$ measured in $\mathcal{M}$ can be written as follows:

$$\text{Len}(x(t)) = \int_0^1 \dot{x}^T H(x) \dot{x} \, dt = \int_0^1 \dot{z}^T J_f^T(z) H(f(z)) J_f(z) \dot{z} \, dt. \tag{1}$$

Since the length of $z(t)$ measured in the Euclidean latent space is $\text{Len}(z(t)) = \int_0^1 \dot{z}^T \dot{z} \, dt$, we can notice that, if $J_f^T(z) H(f(z)) J_f(z) = I$ for all $z \in \mathcal{Z}$, then the distances are preserved, i.e., $\text{Len}(x(t)) = \text{Len}(z(t))$, for all smooth curves $z(t)$. In this respect, a mapping $f$ that satisfies the above condition is considered as a geometry-preserving mapping and called an isometry.

In autoencoder applications, Lee et al. (2022b) showed that the scale does not need to be preserved (i.e., relative distances need to be preserved than the exact distance values) and it is better to search for a *scaled isometry*. A mapping $f$ is called a scaled isometry if $J_f^T(z) H(f(z)) J_f(z) = cI$ for some positive scalar $c$ and for all $z \in \mathcal{Z}$. Or equivalently, all eigenvalues of $J_f^T H J$ should be equal to $c$ for all $z \in \mathcal{Z}$. A family of coordinate-invariant functionals of $f$ that measure how far $f$ is from being a scaled isometry – which are called the *relaxed distortion measures* – have been formulated.

These distortion measures are defined by taking the global expectation of local distortion measures. In practice, rather than being defined by integration over the entire latent space area, they are defined by focusing on the area where data exist. This is achieved by using the latent space probability measure $P_g$ within $\mathcal{Z}$, which is the push-forward of the data distribution in $\mathcal{X}$ by the encoder $g$.

From a practical computational standpoint, one of the most convenient relaxed distortion measures is

$$\mathcal{F}(f; P_g) = \frac{\mathbb{E}_{z \sim P_g}[\text{Tr}((J_f(z)^T H(f(z)) J_f(z))^2)]}{\mathbb{E}_{z \sim P_g}[\text{Tr}(J_f(z)^T H(f(z)) J_f(z))]^2} = \frac{\mathbb{E}_{z \sim P_g}[\sum_i \lambda_i^2(z)]}{\mathbb{E}_{z \sim P_g}[\sum_i \lambda_i(z)]^2}, \tag{2}$$

where $\lambda_i(z)$ are the eigenvalues of $J_f(z)^T H(f(z)) J_f(z)$. In practice given a set of data points $\{x_i \in \mathcal{X}\}_{i=1}^N$, the sampling $z \sim P_g$ is performed by $z = g(x_i)$ for $i \sim \text{Uniform}(1, \dots, N)$. This measure is greater or equal to zero, and is zero if and only if $f$ is any scaled isometry (with respect to $P_g$). Moreover, $\mathcal{F}$ exhibits the desired scale-invariant property, meaning it does not inherently favor any specific scale of $J_f^T H J_f$.

---

[1]If $f$ is a smooth injective mapping and the Jacobian $f$ is full rank everywhere, then the image of $f$ is a differentiable manifold.

In the isometric regularization of autoencoders, the loss function is defined as the sum of the reconstruction loss and the relaxed distortion measure for learning a scaled isometric decoder:

$$\min_{f,g} \frac{1}{N} \sum_{i=1}^{N} \|f(g(x_i)) - x_i\|^2 + \alpha \mathcal{F}(f, P_g),\tag{3}$$

where $\alpha$ is the weight parameter or regularization coefficient. We refer to autoencoders trained with the above loss as the **Isometrically Regularized Autoencoders (IRAEs)**.

In practice, to avoid the computation of the full Jacobian $J_f$ that requires sufficient memory and computational cost, we use Hutchinson's stochastic trace estimator (Hutchinson, 1989) and only need to compute the Jacobian-vector and vector-Jacobian products. Additionally, to extend the measure's influence to encompass the region between the data points, the latent distribution $P_g$ is augmented by $z = \delta z_1 + (1 - \delta)z_2$ such that $z_i \sim P_g, i = 1, 2$, where $\delta$ is uniformly sampled from $[-0.2, 1.2]$ (Chen et al., 2020a; Lee et al., 2022b).

In (Lee et al., 2022b), the ambient space Riemannian metric $H(x)$ is assumed to be the identity $I_D \in \mathbb{R}^{D \times D}$ for all $x \in \mathcal{X}$. In this paper, we go beyond the identity metric assumption and explore diverse choices of $H(x)$, which will be discussed in the following section. A simple toy example that illustrates the effect of $H(x)$ in isometric representation learning is provided in Appendix A.

## 3 RIEMANNIAN METRICS IN THE AMBIENT DATA SPACE

In this section, we first review an unsupervised metric learning approach, the inverse density-based metric proposed in (Arvanitidis et al., 2020). Then we propose a neural feature-based metric constructed by transferring knowledge from a pre-trained feature extraction model.

### 3.1 INVERSE DENSITY-BASED METRIC

Adopting (Arvanitidis et al., 2020), we consider an unsupervised diagonal metric of the following form:

$$H(x) = (\epsilon \cdot p_{\mathcal{X}}(x) + \eta)^{-1} \cdot I_D,\tag{4}$$

where $p_{\mathcal{X}} : \mathcal{X} \to \mathbb{R}_{>0}$ is the probability density function, and $\epsilon, \eta > 0$ are scaling factors to lower and upper bound the metric, respectively. Through its construction, low-density regions are allocated greater volumes, resulting in increased distances between distinct clusters. In this work, we use the Gaussian Mixture Model (GMM) to approximately fit $p_{\mathcal{X}}$. Using more sophisticated models for density estimation, such as normalizing flows (Dinh et al., 2016) and diffusion models (Ho et al., 2020), might lead to enhanced performance.

### 3.2 NEURAL FEATURE-BASED METRIC

In this section, we propose a neural feature-based metric that can leverage knowledge from any pre-trained neural network model. Denote a pre-trained feature extractor by $F : \mathcal{X} \to \mathbb{R}^K$, where we assume $m < K$. We denote the Jacobian of $F$ by $J_F(x) \in \mathbb{R}^{K \times D}$. The model $F$ can be any feature extraction model. We define the neural feature-based metric as the pullback of the Euclidean metric in the feature space $\mathbb{R}^K$ to $\mathcal{X}$ as follows:

$$H(x) := J_F^T(x)J_F(x).\tag{5}$$

First of all, this metric is positive semi-definite, but cannot be positive-definite if $K < D$, which is typically the case in most situations. Therefore, $H$ may not be a valid Riemannian

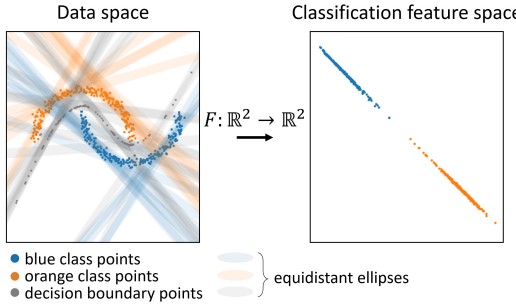

Figure 2: An illustrative example of neural feature-based metrics. Each ellipse represents the set of equidistant points from a center point $x$ given the neural feature-based metric $H(x)$.

metric for many cases. Nevertheless, since our interest is the geometry of the $m$-dimensional embedded manifold $\mathcal{M}$, the projected metric to $\mathcal{M}$ can be a valid metric since we assume $m < K$. Specifically, the projected metric to $\mathcal{M} = f(\mathcal{Z})$ expressed in $\mathcal{Z}$ is written as follows: $J_{F \circ f}(z)^T J_{F \circ f}(z)$. This is an $m \times m$ positive-definite matrix if the Jacobian of the composition $J_{F \circ f}(z) \in \mathbb{R}^{K \times m}$ is full rank. In practice, it is recommended to verify if the rank of $J_F$ is not too low, i.e., if $F$ does not contract the input space too much to a very low-dimensional manifold.

We provide a simple illustrative example of the neural feature-based metrics in Figure 2, using a pre-trained classifier. Suppose we are given a classifier for the two-moon dataset, where the classification feature extractor $F : \mathbb{R}^2 \to \mathbb{R}^2$ maps orange and blue points in the left figure to the points in the right figure, making the orange and blue points linearly separable. The neural feature-based metrics $H(x)$ for this classification feature extraction model $F$ evaluated at randomly sampled orange points, blue points, and gray decision boundary points are visualized as equidistant ellipses in the left figure, where the equidistant ellipse at $x$ is defined to be $\{\bar{x} \in \mathbb{R}^2 \mid (\bar{x} - x)^T H(x)(\bar{x} - x) = 1\}$. This neural feature-based metric defines a geometry where points of the same class become very close to each other, while points of different classes become distant. As one can notice from the figure, the ellipses are very long and narrow, implying that $H(x)$ is close to being semi-definite.

In the subsequent sections, we focus on self-supervised and transfer learning scenarios. In self-supervised learning, given a set of pre-defined semantic-preserving transformations, $F$ is trained in a way that two transformed versions of the same data have similar features while different data have dissimilar features (Chen et al., 2020b; Tian et al., 2020; Caron et al., 2020; Grill et al., 2020; Bardes et al., 2021). In transfer learning, $F$ is pre-trained with a large-scale dataset – which is much bigger in size and diversity than the dataset we will use for manifold representation learning – for some auxiliary task (e.g. classification) (Simonyan & Zisserman, 2014; Szegedy et al., 2016; Deng et al., 2009).

## 4 EXPERIMENTS

We use the standard benchmark image datasets such as the *MNIST* (Deng, 2012), *Fashion MNIST* (Xiao et al., 2017), and *CIFAR10* (Krizhevsky et al.) due to the availability of large-scale pre-trained models and self-supervised learning techniques specifically designed for them. However, it is essential to note that our general framework is not restricted to image datasets only. Implementation details such as network architectures can be found in Appendix B. And, additional experimental results are given in Appendix C. First, we provide how the representations of IRAEs change as the regularization coefficient gradually increases. Second, we show how our isometric regularization with various ambient space metrics influences the representations of Variational Autoencoder (VAE) Kingma & Welling (2013).

### 4.1 MNIST AND FASHION MNIST: TWO- AND THREE-DIMENSIONAL MANIFOLD ANALYSIS

In this section, we focus on the MNIST and Fashion MNIST image datasets, exploring two-dimensional and three-dimensional latent space cases.

#### 4.1.1 GEOMETRY OF INVERSE DENSITY AND NEURAL FEATURE-BASED METRICS

We qualitatively compare the identity metric, the inverse density-based metric, and various neural feature-based metrics. For the inverse density-based metric, which we denote by *invden*, we use the GMM with 10 components with spherical covariance type for $p_{\mathcal{X}}$. For neural feature-based metrics, we use ImageNet pre-trained neural networks that have various parameter numbers: (i) *MNASNet* (Tan et al., 2019) (2.2M), (ii) *ResNet34* (He et al., 2016) (21.8M), (iii) *ConvNeXt* (Liu et al., 2022) (88.6M), and (iv) *VGG16* (Simonyan & Zisserman, 2014) (138.4M), each of which is denoted by *nf-mnasnet*, *nf-resnet34*, *nf-convnext*, and *nf-vgg16*, respectively. Also, we adopt the self-supervised learning method, SimCLR (Chen et al., 2020b), and train the ResNet18 (11.7M) for MNIST and Fashion MNIST datasets, separately. We denote them by *nf-simclr*. For similarity transformations, we use the horizontal flip, crop and resize, and Gaussian blur.

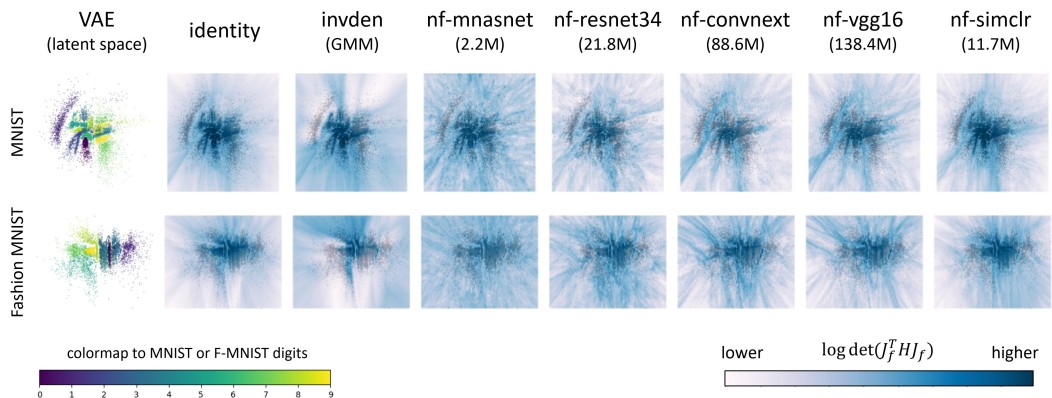

Figure 3: The $\log \det(J_f^T H J_f)$ functions of $z \in \mathbb{R}^2$ are visualized in the VAE latent spaces, where the ambient space metric $H(x)$ is either the identity, inverse density-based, or neural feature-based metrics. The bluer regions indicate higher areas in each actual Riemannian manifold, implying that they are more contracted in the latent space.

Since the data space metrics $H$ are defined in very high-dimensional ambient data space $\mathbb{R}^D$, e.g., $D = 784$ for MNSIT and Fashion MNIST, it is challenging to directly analyze and compare them. Instead, we compare the pull-back metrics in the two-dimensional latent space of the pre-trained Variational Autoencoder (VAE), i.e., denoting the decoder by $f$, we compare $J_f^T(z)H(f(z))J_f(z) \in \mathbb{R}^{2\times2}$ for $z \in \mathbb{R}^2$. Here, we use VAE decoder since AE tends to produce severely distorted representations, making it difficult to recognize distributions of the pull-back metrics.

The $\log \det(J_f^T H J_f)$ is visualized as a function of $z$ in Figure 3; the bluer, the higher. The determinant at $z$ given each metric $H$ measures the local area at $f(z)$ in each Riemannian manifold. In other words, the actual areas of the blue regions in the manifold are bigger than what is shown in the latent spaces. Compared to the identity metric cases, the inverse density-based metrics produce whiter regions in higher-density regions as expected. The desired correlation between density and color, i.e., higher density corresponds to a whiter color, might appear less precise, due to the use of a simple ambient space density model GMM.

While neural feature-based metrics with fewer parameters, such as nf-mnasnet and nf-resnet34, lack distinct patterns, those with a greater number of parameters, like nf-convnext and nf-vgg16, reveal blue bands that separate classes. The presence of these blue bands indicates that data from different classes are indeed more distantly positioned in the manifolds than what is visualized in the latent spaces. The self-supervised neural feature-based metric, nf-simclr, exhibits comparable patterns to nf-convnext and nf-vgg16, despite having a significantly lower number of parameters.

In the subsequent sections, we compare the effects of the identity, invden, nf-vgg16 – since it has more clear patterns than the other smaller pre-trained models –, and nf-simclr metrics in isometric representation learning.

### 4.1.2 QUALITATIVE ANALYSIS

We train the AE and IRAEs with the identity, inverse density (invden), and neural feature metrics (nf-vgg16 and nf-simclr), with the MNIST and Fashion MNIST data by using two and three-dimensional latent spaces. Figure 4 and Figure 5 show their two- and three-dimensional latent space representations, respectively, with class labels indicated by different colors.

Overall, the AE exhibits results lacking clear patterns. For IRAE, depending on the metric, the resulting characteristics are very different. Compared to the identity metric, the invden metric results in slightly more empty spaces between classes, since it expands low-density regions. The IRAEs with the neural feature metrics produce more semantic-aware-representations, demonstrating more apparent clustering structures.

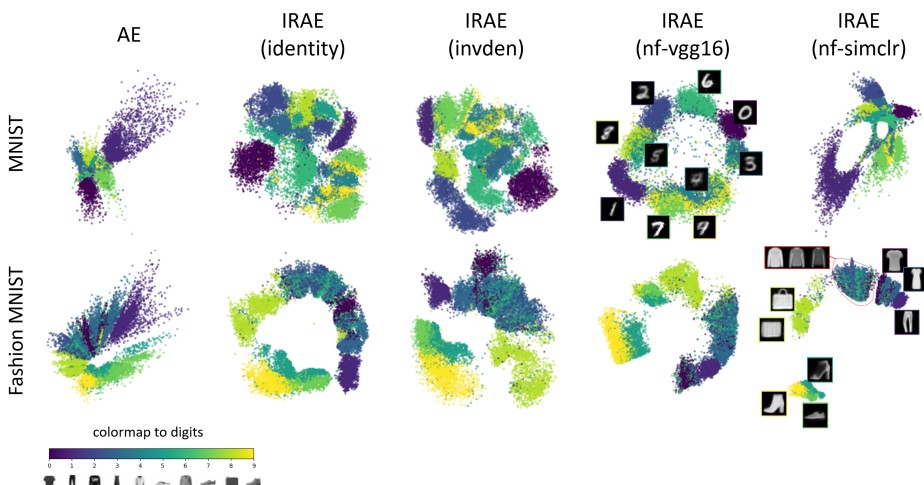

Figure 4: Two-dimensional latent space representations of the MNIST (upper) and Fashion MNIST (lower) datasets. Four different isometric representations are obtained with four different ambient space metrics. Images shown in the figures are generated by using the trained decoders.

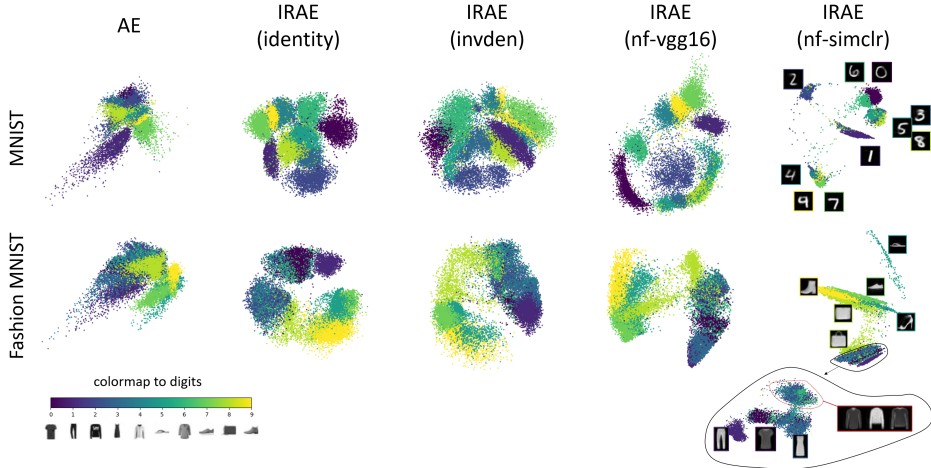

Figure 5: Three-dimensional latent space representations of the MNIST (upper) and Fashion MNIST (lower) datasets. Images shown in the figures are generated by using the trained decoders. HTML files for interactive 3D plots can be downloaded: MNIST and F-MNIST.

In Figure 4 (*Upper*) for two-dimensional MNIST latent spaces, the nf-vgg16 metric results in a donut-shaped distribution, where digits with locally curvy features (e.g., 2, 6, 0, 3) are mainly located on the upper right, whereas digits with less curvature features (e.g., 1, 7) are located on the lower left. And the digits with both features (e.g., 8, 5, 4, 9) are located in between them. The nf-simclr metric produces more dramatic patterns, where the distribution is largely divided into 3 clusters, each of which consists of the following classes: (i) 1, (ii) 4, 7, 9, and (iii) 0, 2, 3, 5, 6, 8.

In Figure 4 (*Lower*) for two-dimensional Fashion MNIST latent spaces, the IRAEs overall produce four clear clusters: (i) 1: Trouser, (ii) 5: Sandal, 7: Sneaker, 9: Ankle boot, (iii) 8: Bag, and (iv) 0: T-shirt, 2: Pullover, 3: Dress, 4: Coat, 6: Shirt. In the fourth cluster, various tops are mixed, implying that the manifold dimension is too small to fit their variability. The nf-simclr metric results in a relatively clearer clustering pattern, where the distance between the shoe and other classes has become very large.

The overall pattern observed in Figure 5 for the three-dimensional latent spaces of MNIST and Fashion MNIST remains consistent with the trends observed in the two-dimensional latent space cases

Table 1: The 2D and 3D latent KNN classification accuracy for MNIST and Fashion MNIST data; the higher, the better. The number of nearest neighbors is indicated by the number in the subscript.

| | | MNIST | | | | | Fashion MNIST | | | | |
|---|---|---|---|---|---|---|---|---|---|---|---|
| | | AE | IRAE (identity) | IRAE (invden) | IRAE (nf-vgg16) | IRAE (nf-simclr) | AE | IRAE (identity) | IRAE (invden) | IRAE (nf-vgg16) | IRAE (nf-simclr) |
| 2D | $ACC_1$ | 77.97 | 78.97 | 79.06 | 80.15 | **81.28** | 68.39 | **69.67** | 69.00 | 69.62 | 69.15 |
| | $ACC_5$ | 82.59 | 83.64 | 83.26 | 84.71 | **85.48** | 73.03 | 74.21 | 74.25 | 74.39 | **74.68** |
| | $ACC_{10}$ | 83.27 | 84.08 | 84.28 | 85.33 | **86.14** | 74.42 | 75.36 | 75.61 | 75.46 | **75.79** |
| | $ACC_{20}$ | 83.79 | 84.74 | 84.81 | 85.71 | **86.29** | 75.02 | 76.31 | **76.55** | 75.88 | 76.20 |
| 3D | $ACC_1$ | 83.61 | 84.81 | 84.85 | **86.73** | 84.91 | 72.01 | 73.21 | **74.35** | 73.03 | 73.96 |
| | $ACC_5$ | 86.87 | 88.91 | 88.87 | **90.17** | 88.01 | 76.53 | 77.50 | **78.88** | 77.26 | 78.20 |
| | $ACC_{10}$ | 87.31 | 89.24 | 89.32 | **90.65** | 88.25 | 77.59 | 78.66 | **79.71** | 78.43 | 79.1 |
| | $ACC_{20}$ | 87.58 | 89.57 | 89.43 | **90.75** | 88.67 | 78.33 | 79.13 | **80.01** | 78.98 | 79.51 |

(interactive 3D plots can be downloaded through the links provided in the figure caption). The neural feature-based metrics produce more semantic-aware representations, where the data with similar semantics are located nearby, compared to the other metrics constructed in a fully unsupervised manner.

Notably, the nf-simclr metrics unveil the hierarchical structures within the datasets, which are not observed in the two-dimensional latent spaces. In the MNIST case, there are five distinct clusters, each of which is comprised of the following digits: (i) 1, (ii) 2, (iii) 4, 9, 7, (iv) 6, 0, and (v) 3, 5, 8. In the Fashion MNIST case, at the top of the hierarchy, there are five clusters: (i) 5: Flat sandal, (ii) 5: High-heel sandal, 7: Sneaker, and 9: Ankle boot, (iii) 8: Bag without a handle, (iv) 8: Bag with a handle, 0: T-shirt, 2: Pullover, 3: Dress, 4: Coat, and 6: Shirt, and (v) 1: Trouser. Interestingly, the same class images are sometimes divided into different clusters based on their visual appearance (e.g., 5: Sandal and 8: Bag). In particular, in the fourth cluster, bags with handles are together with various tops, which seems because the rounded portion of the bag handle is visually similar to the neck area of the tops. The fourth cluster is further divided into four different sub-clusters: (iv-1) 8: Bag with a handle, (iv-2) 0: T-shirt, (iv-3) 3: Dress, and (iv-4) 2: Pullover, 4: Coat, and 6: Shirt, where distinguishing between the pullover, coat, and shirt appears to be challenging when represented in the three-dimensional latent space.

### 4.1.3 QUANTITATIVE ANALYSIS

Table 1 shows the accuracy of the KNN classification performed in the latent spaces, where $k \in \{1, 5, 10, 20\}$ and the number of training and test data is 60000 and 10000, respectively. Generally, the IRAEs produce higher accuracy than AE. In the MNIST case, the IRAEs with the neural feature-based metrics produce the highest accuracy, whereas, in the Fashion MNIST case, the IRAEs with the invden and nf-simclr metrics produce the highest accuracy. This shows that, without using the class labels in the training of autoencoders, isometric regularization can produce representations where data of the same class are closer to each other than data of different classes.

### 4.2 CIFAR10: HIGHER-DIMENSIONAL MANIFOLD ANALYSIS

In this section, we study the effects of various Riemannian metrics in isometric representation learning using the CIFAR10 image dataset. Similar to the MNIST and Fashion MNIST cases, we compare the identity, invden, nf-vgg16, and nf-simclr metrics. We use GMM with ten components for the invden metric, and for the nf-simclr metric, we use the horizontal flip, crop and resize, color jittering, and random grayscale image transformations as semantic-preserving transformations. The variability in CIFAR10 image dataset is too complex to fit with two- or three-dimensional manifolds; we assume 128-dimensional manifolds.

To visualize 128-dimensional latent space representations, we use the UMAP (McInnes et al., 2018) algorithms and map latent vectors in two-dimensional spaces, as shown in Figure 6. All ten classes of data are used in Figure (*Upper*), and the airplane, bird, and truck classes of data are used in Figure (*Lower*). The rightmost figure shows the results of UMAP algorithms applied directly to the image pixel space. We fit a Gaussian distribution for each class and visualize its contour as concentric ellipses. In the upper figure, while complete separability is not achieved among all ten classes, the

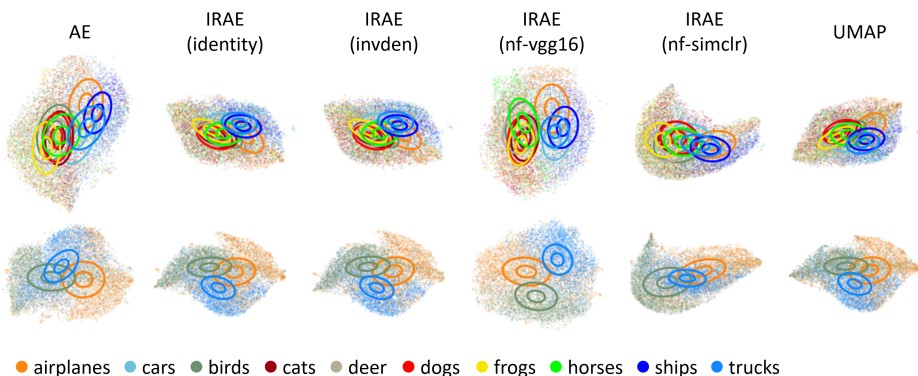

airplanes ● cars ● birds ● cats ● deer ● dogs ● frogs ● horses ● ships ● trucks

Figure 6: Two-dimensional UMAP representations of 128-dimensional latent space representations of the CIFAR10 datasets. *Upper*: The entire set of ten data classes is used; *Lower*: Data from the airplane, bird, and truck classes are employed.

data from vehicle classes – which consist of airplanes, cars, ships, and trucks – and animal classes – which consist of birds, cats, deer, dogs, frogs, and horses – are subtly positioned apart from each other; this feature is the most evident in the nf-vgg16 metric case. In the lower figure, we can also notice that the nf-vgg16 metric results in the best separability.

Table 2 shows the latent space KNN classification accuracy, where $k \in \{1, 5, 10, 20\}$ and the number of training and test data is 50000 and 10000, respectively. The classification accuracy shows a gradual improvement from AE, IRAE with identity, invden, nf-vgg16, to nf-simclr metrics. An intriguing observation is that, as the value of $k$ increases in AE, the performance experiences a notable decline. On the other hand, in IRAE with identity and invden metrics, the performance remains relatively consistent. On the contrary, in IRAE with

Table 2: The KNN classification accuracies in 128-dimensional latent spaces for CIFAR10 data; the higher, the better. The numbers of nearest neighbors used in KNN classification are indicated by the numbers in subscripts.

|  | AE | IRAE (identity) | IRAE (invden) | IRAE (nf-vgg16) | IRAE (nf-simclr) |
|---|---|---|---|---|---|
| $ACC_1$ | 37.04 | 39.31 | 40.24 | 41.46 | **44.56** |
| $ACC_5$ | 33.42 | 38.79 | 40.09 | 42.88 | **45.94** |
| $ACC_{10}$ | 32.77 | 39.58 | 39.74 | 44.77 | **46.54** |
| $ACC_{20}$ | 31.27 | 39.17 | 39.37 | 46.26 | **47.53** |

neural feature metrics, there is a noticeable improvement in performance as $k$ increases. For IRAE with unsupervised identity and invden metrics, there's a performance boost of approximately 2 to 3% over AE. Meanwhile, IRAE with neural feature metrics exhibits an improvement of 9 to 10%.

## 5  DISCUSSION AND CONCLUSION

For the first time, we have conducted a study on how ambient space Riemannian metrics influence isometric representation learning. We have proposed a new family of ambient space metrics, neural feature-based metrics, which can capture the semantics of the data. These metrics exploit knowledge from pre-trained feature extraction models.

Through extensive experiments with standard benchmark image datasets, our neural feature metrics constructed via self-supervised and transfer learning have been compared to the metrics constructed in a fully unsupervised manner, the identity and inverse density-based metric. Our findings demonstrate that the neural feature metrics result in representations that are more attuned to the semantics of the data. They enable data with similar semantics to cluster together, even without explicit labels, sometimes revealing unknown hierarchical structures within the datasets.

Our research highlights the significant impact that the choice of ambient space metrics can have on the resultant isometric representations. Currently, we are in the early stages of exploring a broader range of metrics and their associated consequences. We anticipate that our discoveries will be instrumental in advancing the development of improved methods for selecting metrics across various applications.

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

# APPENDIX

## A    THE EFFECT OF $H(x)$ IN IRAE: TOY EXAMPLE

In this section, we show a simple example that illustrates the effect of the ambient space Riemannian metric $H(x)$ in isometric representation learning. Consider a set of three-dimensional data points lying in a two-dimensional S-curve manifold as shown in Figure 7 (*Left*). We train the vanilla Autoencoder (AE) and Isometrically Regularized Autoencoders (IRAEs) with two-dimensional latent spaces. To apply IRAEs, we need to determine the ambient space metric $H(x) \in \mathbb{R}^{3 \times 3}$ in prior. Here, we compare the identity metric, i.e., $H(x) = I_3$, and a metric that increases quadratically in the $x_3$ direction $H(x) = \frac{1+x_3^2}{20} \cdot I_3$.

Figure 7 shows the resulting two-dimensional latent representations. The AE produces arbitrarily distorted representations. The IRAE with the identity metric produces square-shape representations with a square hole, which preserves the (Euclidean) geometry of the original S-shape manifold – where the height is 10, the length of the S-curve is 10, and it has a square hole. The IRAE with the quadratically increasing metric produces representations where the yellow area expands and the purple area contracts. This is because the ambient space metric $H$ assigns large volumes to regions with higher $x_3$, i.e., the yellow regions, and small volumes to regions with lower $x_3$, i.e., the purple regions. As shown, the metric selection has a significant impact on the final isometric representation.

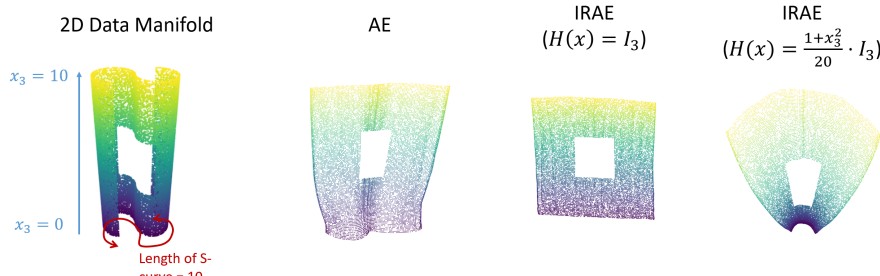

Figure 7: *Left*: A set of three-dimensional data points lying in a two-dimensional S-curve manifold. The color represents $x_3$ value which ranges from 0 to 10, and the length of the S-curve in $(x_1, x_2)$-plane is 10. *Right*: Two-dimensional latent representations of the autoencoder and isometrically regularized autoencoders with different ambient space metrics $H(x)$.

## B    EXPERIMENTAL DETAILS

### B.1    DETAILS ON AMBIENT SPACE METRICS

For the invden metric construction – where $H(x) = (\epsilon \cdot p_{\mathcal{X}}(x) + \eta)^{-1} \cdot I_D$ –, we use the Gaussian mixture model to fit the ambient space density function $p_{\mathcal{X}}(x)$. The number of mixture components is set to be equal to the number of classes (in our experiments, it is 10) and we use the spherical covariance type in GMM to prevent overfitting.

For the pre-trained neural feature metrics, we use weights from the torchvision library. Specifically, we use the MNASNET0-5-Weights.IMAGENET1K-V1 for the nf-mnasnet metric, ResNet34-Weights.IMAGENET1K-V1 for the nf-resnet34 metric, ConvNeXt-Base-Weights.IMAGENET1K-V1 for the nf-convnext metric, and VGG16-Weights.DEFAULT for the nf-vgg16 metric.

For the nf-simclr metric, we train SimCLR models that consist of (i) ResNet18 and (ii) two-layer MLPs (512-512-128) with ReLU activation functions. The 512-dimensional ResNet18 features are used for the metric construction. For MNIST and Fashion-MNIST, the semantic-preserving image transformations are random horizontal flip with probability 0.5, random resize and crop with size 28, and Gaussian blur with kernel size 3. For CIFAR10, the transformations are random horizontal flip with probability 0.5, random resize and crop with size 32, random color jitter with (bright-

ness, contrast, saturation, hue) = (0.4, 0.4, 0.4, 0.1) and probability 0.8, and random grayscale with probability 0.2.

### B.2 NEURAL NETWORK ARCHITECTURES

For MNIST and Fashion-MNIST, we use fully conn ected neural networks for the encoder and decoder. The encoder and decoder are five-layer MLPs with ReLU activation functions, either having the sequence of node numbers 784-256-256-256-256-$z_{\text{dim}}$ or $z_{\text{dim}}$-256-256-256-256-784. The output activation functions are linear and sigmoid for the encoder and decoder, respectively. For CIFAR10, the encoders are convolutional neural networks with the following architecture: i) Conv2d (3, 32, 4, 2, 0), ii) Conv2d (32, 64, 4, 2, 0), iii) Conv2d (64, 128, 4, 2, 0), iv) Conv2d (128, 256, 2, 2, 0), v) Conv2d (256, $z_{\text{dim}}$, 1, 1, 0) with ReLU hidden layer activation functions, and decoders are Convolutional Neural Networks with the following architecture: i) ConvT2d ($z_{\text{dim}}$, 256, 8, 1, 0), ii) ConvT2d (256, 128, 4, 2, 1), iii) ConvT2d (128, 64, 4, 2, 1), iv) ConvT2d (64, 3, 1, 1, 0) with ReLU hidden layer activation functions. The output activation functions are linear and sigmoid for the encoder and decoder, respectively.

### B.3 HYPERPARAMETERS AND TRAINING DETAILS

For all datasets and autoencoders, the batch size is 100, the number of training epochs is 300, and the learning rate is 0.0001. We use Adam optimizer with default parameter settings. For IRAEs, we search for the regularization coefficient in $\{0.000001, 0.00001, 0.0001, 0.001, 0.01, 0.1, 1, 10, 100\}$. For IRAE with the invden metric, we set $\eta = 1$ and additionally search for the $\log \epsilon$ in $\{0.000001, 0.00001, 0.0001, 0.001, 0.01\}$. As the regularization coefficient increases, more isometric representation can be found but the reconstruction error could increase, meaning there is a natural trade-off; see Section C.1. We use different models for visualization and KNN classification (the one that shows the highest classification accuracy does not always produce representations that have prominent visual characteristics). When reporting the KNN classification results, the best regularization coefficient and other hyperparameters are selected based on the KNN classification score. When visualizing the latent space representations, we choose higher regularization coefficients – therefore resulting in more isometric representations – to give better intuitions on the effects of the Riemannian metrics in isometric representation learning. Section C.1 provides how the representations change as the regularization coefficient changes.

## C ADDITIONAL RESULTS

### C.1 REGULARIZATION COEFFICIENT SWAPPING

The regularization coefficient $\alpha$ has a significant impact on the resultant isometric representations. As the coefficient $\alpha$ increases, the more isometric representations can be obtained; however, this often comes at the expense of an increased reconstruction error. Figure 8 and 9 show how the isometric representations with various ambient space metrics change as the regularization coefficient gradually increases, using MNIST and Fashion MNIST datasets. As expected, models trained with too small $\alpha$ produce similar results to the vanilla AEs, whereas models trained with too big $\alpha$ produce representations where data of different classes collapse to each other due to high reconstruction errors. Selecting a proper value of $\alpha$ is important to balance between isometric representation and reconstruction.

### C.2 ISOMETRIC REGULARIZATION IN VAE

For Variational Autoencoder (VAE) Kingma & Welling (2013), our isometric regularization can be straightforwardly added to the VAE loss as well, leading to the Isometrically Regularized VAE (IR-VAE) Lee et al. (2022b). Here, we compare IRVAEs with various ambient space metrics. Figure 10 shows the 2D latent representations obtained by IRVAEs with various ambient space metrics, using MNIST and Fashion MNIST datasets. Unlike the IRAEs' representations, the neural feature-based metrics do not produce representations that capture prominent semantic features. This is because the KL divergence term in VAE ELBO loss that enforces the latent distribution to be Gaussian competes with the isometric regularization term, resulting in less isometric representations.

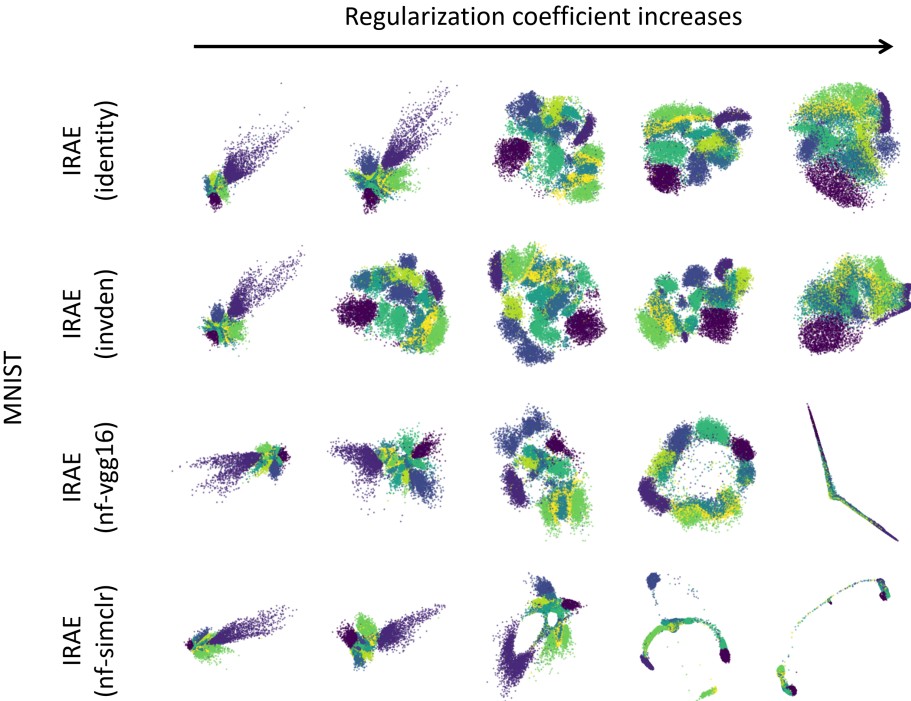

Figure 8: 2D MNIST latent representation obtained by IRAEs with increasing regularization coefficients.

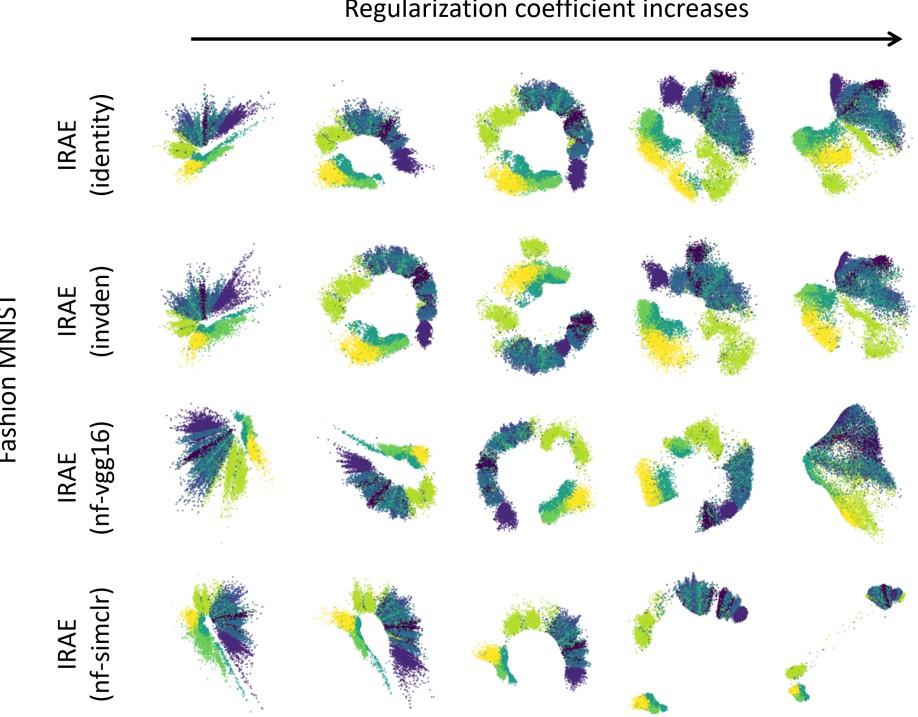

Figure 9: 2D F-MNIST latent representation obtained by IRAEs with increasing regularization coefficients.

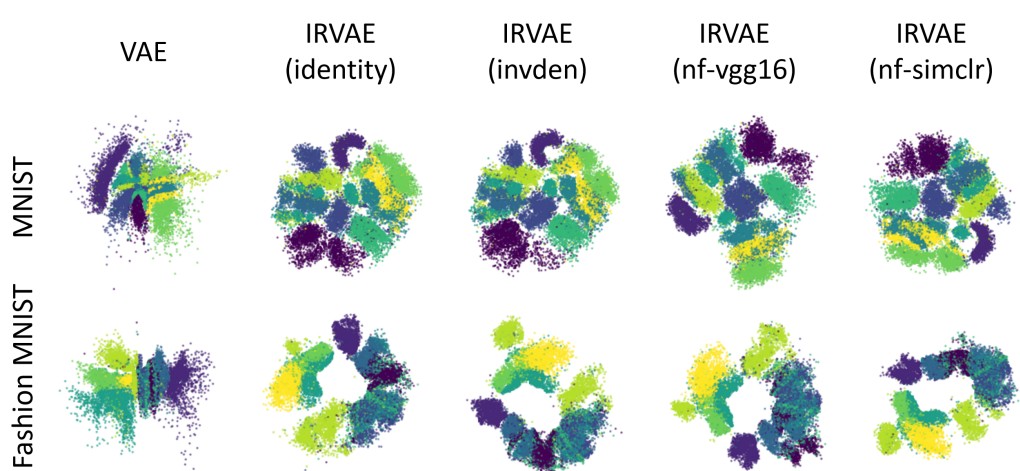

Figure 10: 2D latent representations obtained by IRVAEs with various ambient space metrics, using MNIST and Fashion MNIST datasets.

