# OpenReview forum: "On the Role of Riemannian Metric in Isometric Representation Learning"
_ICLR.cc/2024/Conference — ICLR 2024 Conference Withdrawn Submission_

### Official Review · Reviewer_HnxU · 2023-10-30

**Soundness:** 3 good
**Presentation:** 3 good
**Contribution:** 2 fair
**Rating:** 3
**Confidence:** 5

**Summary:**

The paper considers pull-back metrics in isometrically regularized VAEs, and ask "which metric should be pulled back?". A series of different metrics over the observation space are compared empirically.

While I support the ambition of the paper, I have some concerns that currently prevent me from recommending acceptance. I hope this feedback can be used constructively to improve the paper as I think the core study has value.

**Strengths:**

The study of pullback metrics in unsupervised learning is important as these have shown value from both theoretical and practical perspectives. The question of *which metric* to pull back has, however, not been given a systematic study, which the present paper attempts to rectify.

The provided visualizations are valuable as they can also help us build intuitions about how different models behave. In particular, I found Fig. 3 to be quite interesting.

**Weaknesses:**

## Major things ##
* The idea of using features from pre-trained neural networks to define pull-back metrics has already been considered: https://openreview.net/forum?id=BJslDBkwG
* I partially object to the KNN experiments. Here the metric induced by a pretrained neural classifier is pulled into the latent space. It is then concluded that this metric is better (for knn) than alternative metrics. I trust that this conclusion is correct, but I do not think it is acceptable to skip a discussion of the simple fact that you are now compared supervised learning to unsupervised learning. The neural network-based metric leverages class information, while the others do not. Obviously, it is going to perform better. To repeat: I think it is fine to perform such experiments, but I do not think it is acceptable to not discuss that the experiment effectly compares unsupervised learning to supervised learning (on a supervised learning task).
* I found Fig 6 to be difficult to conclude anything from. To me it looks like all the tried approaches failed.
* The main contribution of the paper is an investigation of the influence of the choice of observation-space metric. Given the experiments, I honestly struggle to draw a conclusion.


## Minor things ##
(I do not need replies to these comments in the rebuttal, but hope the authors will rectify the highlighted issues)
* The term 'isometric' is defined very late in the paper dispite the term being used quite a lot (it's even in the title).
* I think it is unfortunate, that it is not mentioned in the paper that we, in general, should not expect to be able to find an isometric representation (such would imply that the manifold is flat). This is the reason for regularizing towards an isometry rather than actually finding one.
* The phrase 'invden' is used on page 2 but not defined until page 5.
* Eq 1 describes *curve energy* not *curve length* (square roots are missing)
* Why is Deng (2012) cited for having proposed MNIST?
* I miss a citation of the paper "Riemannian metric learning" from Guy Lebanon (which is one of the first papers in the field).

**Questions:**

* On page 5 it is recommended to check the rank of the metric. How do I do this in practice?
* How stable are the produced plots? Specifically, if I were to rerun the code to produce Fig 4 using different random seeds, would I get similar results?

---

### Official Review · Reviewer_7V9h · 2023-10-30

**Soundness:** 2 fair
**Presentation:** 3 good
**Contribution:** 2 fair
**Rating:** 3
**Confidence:** 3

**Summary:**

This paper focuses on Isometric Representation Learning in the autoencoder framework with an emphasis on studying the role of the Riemannian metrics that define the notion of distance on the manifold. The authors build on the narrative from many previous works in this area and explore 3 choices of the metric: Identity,  Inverse Density [Arvanitidis et al., 2020], and then propose a new neural feature-based metric. The neural feature-based metric is basically constructed from the quadratic form from the Jacobian (Eq 5) of a pre-trained network in the task (e.g. ResNet34, VGG16, etc). The expectation is that a more semantically aware metric leads to simultaneously: (1.) final representations being semantically meaningful and structured/related (2.) An isometric latent space built with this metric allows for geometric consistency, i.e. geodesics get mapped to geodesics, etc. Experiments are reported on the MNIST, FMNIST, and the CFAR10 dataset. The overall message of the experiments is to show the effectiveness of the neural feature-based metric, leading to somewhat more meaningful visualizations of data.

**Strengths:**

- Overall, this paper does pose some very interesting questions with regard to regularisation and geometry in autoencoders. The emphasis on choosing different metrics in isometric representation learning seems interesting. On its own, the idea of the neural feature-based metric is also quite clever.
- Generally, the paper is compiled well and the motivating arguments are posed nicely (especially in Figure 2 and Section 3.2)

**Weaknesses:**

- At a high level, I feel somewhat underwhelmed by the conceptual contribution of this paper. It is not surprising that using a pre-trained feature extractor (encapsulating semantic information from data) to build a local metric in Equation 5, leads to a semantically meaningful representation since the feature extractor has done precisely that. It somewhat feels like a chicken and egg problem (using pre-trained representations to build a metric to get new representations (using IRVAE). As of now, there is little evidence that IRVAE using feature-based metrics is convincingly superior to the original features themselves)
- To this aid, an aspect that is not explored in this paper is the role of enforcing isometry to get good representations. It would be more convincing to have the qualitative and quantitative analysis comparing features before and after enforcing the isometry condition. i.e. how semantically meaningful and isometric are the pre-trained feature extractors already (i.e. before constructing the metric and training the IRVAE)? Does the metric construction and the resultant IRVAE lead to any additional benefit? I suspect that there will be a violation of the geometric structure in these pre-trained representations, but this needs to be elaborated on more concretely in the experiments. I am especially curious if any demonstration can be made that highlights distortion in geodesics?
- Another potential avenue that feels missing is the role of the amount of training data. It is interesting to know: can the neural feature-based metrics be trained with fewer data? How would Tables 1 & 2 change in that case? A plot of performance vs % training data would be very illuminating demonstrating the power of choosing each metric.

**Questions:**

Minor Comments

- invden [Arvanitidis et al., 2020] used on page 2 before its formal description on  page 5
- In Figure 3, is the blue-white color map *exactly* the same for all the plots?
- All captions in the figures could benefit immensely by a sentence or so clearly highlighting the message of that experiment.
- Is Equation 2 correct? I could not trace how it can be zero in the case of scaled isometry, shouldn’t that yield all eigenvalues to be 0? Given the importance of this equation for the narrative of the paper, this typo is unfortunate.

Overall:
This paper attempts an interesting study on the role of metrics in isometric representation learning. While I appreciate and credit the authors for the angle of investigation, I am not yet convinced that the conceptual contributions are enough for acceptance to ICLR.

---

### Official Review · Reviewer_7zcn · 2023-10-31

**Soundness:** 2 fair
**Presentation:** 2 fair
**Contribution:** 1 poor
**Rating:** 3
**Confidence:** 5

**Summary:**

This paper proposes a method for constructing neural feature-based metrics capable of capturing data semantics by adopting knowledge from any pre-trained feature extraction model. The proposed metric was compared with the identity metric and the inverse density-based metric on MNIST, Fashion MNIST, and CIFAR10 datasets.

**Strengths:**

The paper addresses an important problem of manifold learning which is identification of metrics considering semantic properties of data.

**Weaknesses:**

There are several theoretical and experimental limitations with the paper as discussed in the questions.

**Questions:**

-	How do you assure that the neural feature-based metric as the pullback? More precisely, the feature extractor does not need to be a smooth map, and its domain/range may not be a smooth manifold. Also, as mentioned in the paper, J may not be full-rank, and it is not in various cases. Indeed, we do not have these properties in the data usually, which makes the problem, i.e. defining a semantically consistent metric, a challenge. The paper’s main novel claim is resolving this challenge, however, the challenge is not clearly explored and solved. Since this is the main claim of the work, these assumptions should be investigated in detail both mathematically/experimentally.

-	Can you extend the analyses using stronger feature extractors on semantically richer datasets, such as Imagenet, Coco, and/or other tasks e.g. for NLP/speech recognition? These analyses are required to validate the assumptions and claims in detail.

-	Experimental analyses for the self-supervised and transfer learning scenarios should be also extended considering different setups for these scenarios.

---

### Official Review · Reviewer_jvXY · 2023-11-06

**Soundness:** 2 fair
**Presentation:** 3 good
**Contribution:** 2 fair
**Rating:** 5
**Confidence:** 4

**Summary:**

This paper proposes a new isometric representation learning method, which adds a projection from pre-trained neural networks. The motivation of this method is to introduce semantic information to isometric representation learning, leading to more discriminative representations.

**Strengths:**

This idea is new. Combining pre-trained models and unsupervised models, I think, may be a significant research direction.

**Weaknesses:**

In the designed methods and experiments, using a big Resnet model for MNIST  and CIFAR datasets seems to be somewhat overkill and overused. A pre-trained VGG-16 and a Resnet-34 model are enough for the MNIST and CIFAR datasets. They already learn very discriminative representations without isometric representation learning. So in this setting, we cannot know the meaning of isometric representation learning. Maybe the isometric representation learning will destroy the discriminative ability of the pre-trained models.

**Questions:**

(1) More experiments are needed to show the benefits of combining isometric representation learning and pre-trained models. Maybe some more complicated experiments are needed, where the ResNet 34 or VGG-16 cannot achieve good performance. In this case, if you can show that combining isometric representation learning with the deep network can achieve good performance, then the experiments can demonstrate the advantages of your method.

(2) What is the application of the method? With the developments of visual foundation models, the representations of images have been learned very well. In this case, I am afraid that the application of this method is limited.

(3) Some resource assumption including the memory and time is beneficial. We can more comprehensively evaluate the meaning of this method.

(4) Maybe some theoretical analysis and derivation are needed to show, why can we design the metric in the format in section 3.2. How does it transfer knowledge from the pre-trained models to the geometry. I know this seems difficult, but it may be important.

---

### Author Response · Authors · 2023-11-20
**Withdrawal of Paper**

Dear Reviewers,

We would like to inform you that, upon thorough deliberation, we have reached the decision to withdraw our paper.
We appreciate the reviewers for acknowledging that the problem we aimed to address is important and interesting.
Simultaneously, we acknowledge the need for additional experimental results.

We wish to express our profound gratitude for the considerable time and effort you have dedicated to the evaluation of our work. The insights and feedback you have provided will play a pivotal role in guiding the trajectory of our research.

Sincerely,